

# Integrating mindfulness and physical activity: a meta-analysis of mindful movement interventions for symptoms of anxiety and depression among university students

Xinjian Xu[1,2], Borhannudin Bin Abdullah[1], Shamsulariffin Bin Samsudin[1] and Yongneng Tan[3]

[1] Faculty of Educational Studies, Universiti Putra Malaysia, Selangor, Malaysia
[2] Faculty of Physical Education, Sanming University, Sanming, Fujian Province, China
[3] Library, Qingdao University of Science and Technology, Qingdao, Shandong Province, China

Corresponding authors
Xinjian Xu, xuxjsmxy@163.com
Borhannudin Bin Abdullah,
borhannudin@upm.edu.my

## ABSTRACT

**Background.** University students are increasingly vulnerable to mental health problems, such as anxiety and depression, which have been associated with academic stress, job market uncertainty, and lifestyle-related factors such as physical inactivity and poor sleep habits. Mindful movement, defined as structured programs integrating mindfulness practice with physical movement, offers a holistic approach to reducing anxiety and depression, though its effectiveness and specific intervention differences—such as duration, delivery mode, and behavior change techniques—remain underexplored. This study evaluated the effects of mindful movement interventions on anxiety and depression among university students.

**Methods.** A systematic search of six databases up to August 2024 identified 19 randomized controlled trials involving a total of 1,697 participants. A random-effects model was employed to pool effect sizes; subgroup analyses were conducted to explore sources of heterogeneity, and sensitivity analyses were performed to test the robustness of the results.

**Results.** The meta-analysis indicated that mindful movement interventions significantly reduced anxiety (standardized mean difference (SMD) $= -0.42$, 95% confidence interval (CI) $[-0.52$ to $-0.31]$, $p < 0.0001$, $I^2 = 70\%$) and depression (Standardized Mean Difference (SMD) $= -0.61$, 95% CI $[-1.01$ to $-0.20]$, $p < 0.0001$, $I^2 = 87\%$) compared to control groups. Subgroup analyses demonstrated no significant differences in effect sizes across different mindful movement types (yoga, tai chi, qigong). Meta-regression indicated that intervention type did not significantly explain heterogeneity ($p = 0.495$). Publication bias was not observed in most subgroups, except for potential bias in the tai chi group due to limited studies.

**Conclusion.** Mindful movement interventions are associated with enhanced mental health among university students, with no significant differences in the effects of yoga, tai chi, and qigong on reducing symptoms of anxiety and depression. It supports treating mindful movement as a unified approach to analyzing its impact on mental health. Residual heterogeneity highlights the need for further research on

specific intervention elements and intervention and participant-related factors. PROS-PERO registration number (CRD42024569238).

## INTRODUCTION

University students worldwide experience high levels of psychological stress, often manifested as symptoms of anxiety and depression. These mental health challenges stem from academic pressures, career uncertainty, lifestyle imbalances, and digital overexposure, and are increasingly recognized as critical public health concerns (*Auerbach et al., 2016*; *Hunt & Eisenberg, 2010*). In China, where nearly 10 million students graduate each year, the growing pressure of academic performance, competitive employment markets, and familial expectations have contributed to rising levels of psychological distress among university students (*Hua & Sun, 2021*).

Research has shown that both physical activity and mindfulness meditation may alleviate such symptoms by promoting emotional regulation and stress reduction (*Galante et al., 2021*; *Hua & Sun, 2021*). Among Chinese university students, notable gender and grade-level differences exist in physical activity participation: male students prefer aerobic and resistance exercises, while female students are more inclined toward light-to-moderate activities such as jogging, yoga and walking (*Othman et al., 2022*). These results highlight the need to further explore integrated interventions combining mindfulness and physical activity for mental health benefits. This raises a critical question: could mindful movement, which integrates physical activity with mindfulness meditation, provide more effective treatment outcomes for university students struggling with anxiety and depression? This possibility warrants further empirical investigation, especially through comprehensive synthesis of existing evidence.

Mindful movement is an integrated approach that combines mindfulness practices with physical activities, fostering individuals' awareness of the connection between body and mind through focused attention on bodily sensations, breathing, and movement (*Clark, Schumann & Mostofsky, 2015*; *Ni et al., 2024*). Unlike mindfulness-based interventions, which involve meditative or cognitive practices without active physical engagement (*Li et al., 2023*; *Reangsing, Abdullahi & Schneider, 2023*), mindful movement interventions emphasize active physical participation, making them particularly appealing for individuals who may not prefer traditional mindfulness techniques. This distinction underscores their unique value in addressing mental health challenges.

Existing meta-analyses have explored various aspects of mental health interventions among university students. For instance, some studies have examined the effects of specific types of physical activity, such as aerobic and resistance exercises, on reducing anxiety and depression (*Guo et al., 2020*; *Lin et al., 2022*). Others have investigated mindfulness-based interventions, a topic that has been extensively analyzed in meta-analyses

(*Li et al., 2023*; *Reangsing, Abdullahi & Schneider, 2023*; *Yosep, Mardhiyah & Sriati, 2023*). However, most of these studies either focus exclusively on mindfulness practices without incorporating physical activity, or examine physical activity interventions in isolation without integrating mindfulness practices. Recent reviews, such as an integrative analysis of interventions combining physical activity and mindfulness components (*Breedvelt et al., 2019*; *Remskar et al., 2022*; *Remskar et al., 2024*), highlight the potential synergistic effects of these combined approaches, emphasizing the need for more targeted evaluations among university populations. Theoretical frameworks like the embodied mindfulness model (*Schuman-Olivier et al., 2020*) offer a conceptual basis for understanding how mindful movement may uniquely impact mental health outcomes.

In addition, certain studies have analyzed the mental health benefits of individual mindful movement practices, such as the effects of yoga and mindfulness meditation on anxiety and stress (*Breedvelt et al., 2019*), the impact of tai chi (TCC) on psychological disorders (*Jiang et al., 2020*), and the benefits of qigong for reducing stress and anxiety (*Wang et al., 2014*). While these contributions provide valuable insights, there remains a significant lack of meta-analyses that systematically and comprehensively examine mindful movement as a holistic intervention. To date, no study has specifically addressed the combined effects of mindful movement interventions on both anxiety and depression among university students.

Mindful movement interventions, which are rooted in Eastern practices such as tai chi and qigong, may offer a culturally acceptable and accessible alternative to conventional therapies. Thus, examining the effectiveness of mindful movement interventions within the Chinese university population is both timely and culturally meaningful. This study defines mindful movement interventions as the combination of mindfulness practice and physical activities. Adopting an integrative perspective, it systematically evaluates the overall effect size of mindful movement interventions, such as yoga, tai chi, and qigong, on anxiety and depression among university students and compares the differences in the impacts of various types of mindful movement interventions on anxiety and depression in this population.

## METHODS

### Research design
This meta-analysis is reported in accordance with the Preferred Reporting Items Systematic Reviews and Meta-Analyses (PRISMA) guidelines (*Page et al., 2021*), and registered the study in PROSPERO (CRD42024569238). All analyses were pre-specified in the PROSPERO registration, and no *post hoc* analyses were conducted.

### Literature search strategy
To identify the target studies, the literature search was independently conducted by two researchers (X.X.J. and B.A.). This study conducted a comprehensive literature search in five major databases, including PubMed, EBSCO, Cochrane Library, Web of Science, and Embase. Additional searches were performed in a Chinese database, CNKI. The search focused on randomized controlled trials (RCTs) examining the effects of mindful

movement interventions on the mental health of university students, particularly depression and anxiety. The search period extended up to August 2024. The search strategy see Table S1.

## Inclusion and exclusion criteria
### Participants
The study adheres to the definitions of depression and anxiety provided by the World Health Organization's International Classification of Diseases. Therefore, the inclusion criteria for this study targeted non-physical education undergraduate students in China who exhibited symptoms of depression or anxiety. Specifically, participants were included if they had clinical diagnoses based on professional assessment, or exhibited subclinical levels of symptoms as determined by validated depression and anxiety scales. Physical education students were excluded because their higher baseline physical activity levels might confound the effects of mindful movement interventions on mental health outcomes.

### Interventions
This study investigates the effects of mindful movement interventions on alleviating symptoms of anxiety symptoms. The included intervention types comprise yoga, tai chi, qigong without referring to other types of mindful movement. The intervention duration is set to a minimum of two weeks, and eligible control groups include those with no physical activity interventions, with or without a waitlist control group.

### Comparator
This study employs a two-arm comparison of mindful movement interventions and control group, and a multi-arm comparison to evaluate the effectiveness of different types of mindful movement interventions.

### Outcome measures
The outcome factors to be analyzed before and after the intervention must include at least one of depression or anxiety, assessed using validated tools or scales designed to measure these conditions.

### Study design
Only randomized controlled trials (RCTs) that are available in either English or Chinese.

### Exclusion criteria
The exclusion criteria for the study were as follows. First, studies employing quasi-experimental designs or single-group pre-post designs were excluded to ensure methodological rigor. Second, theses, conference proceedings, or abstracts were excluded if the full text and data was not accessible.

## Data extraction
Two investigators, the first author (X.X.J) and the corresponding author (B.A.), independently screened all included studies at least twice. Any disagreements were resolved through discussion, and if needed, another researcher (S.S) was consulted for the final decision. One researcher (TYN) was specifically responsible for organizing incomplete

data and contacting the original authors for further information. Initially, all studies were preliminarily screened based on their titles and abstracts. After removing duplicates and irrelevant articles, the remaining studies were thoroughly reviewed in full to determine their final eligibility for inclusion. Data extraction focused on the following key elements: first author, publication year, intervention participants (sample size, gender, and age of each group), intervention details (type of exercise, duration, and frequency), and outcome measures. For the identification of further related publications, we also retrieved gray literature from Opengrey (https://opengrey.eu/).

### Risk of bias and quality assessment

Two investigators independently assessed the risk of bias (ROB) in the included studies using the Cochrane Handbook's tool for evaluating the risk of bias in randomized controlled trials (RCTs) (*Higgins et al., 2011*). The assessment covered five domains: the randomization process, intervention bias, attrition bias, outcome measurement bias, and other sources of bias. Blinding of participants and personnel was assessed separately, acknowledging that blinding is often not feasible in behavioral intervention studies; any associated risk was reported and discussed rather than excluded. Any disagreements were resolved through discussion, and a third researcher was consulted for arbitration if necessary. We summatively rated the overall ROB of each study as follows: studies were classified as having low ROB if none of the domains above was rated as high ROB and three or less were rated as unclear risk, and as moderate ROB if one was rated as high ROB or none was rated as high ROB, but four or more were rated as unclear risk, and all other cases were assumed to pertain to high ROB (*Cipriani et al., 2018*).

### Statistical analyses

To minimize differences in interventions, we used the mean and standard deviation (SD) of effect sizes before and after the intervention for data synthesis. Statistical analyses, including sensitivity analysis, subgroup analysis, heterogeneity testing, meta-regression, and publication bias assessment (Begg/Egger's test), were conducted using Stata 18.0 software. A random-effects model was applied to pool the effect sizes of mindful movement interventions on depression and anxiety, with standardized mean differences (SMDs; specifically, Hedges' g) and 95% confidence intervals (CIs) as the effect size metrics. SMDs were interpreted as small (0.2), medium (0.5), and large (0.8) effects (*Le Fevre, 2005*). We assessed publication bias through visual inspection of funnel plots and statistical tests, including Begg's rank correlation test. We evaluated the certainty of the evidence using the GRADE approach.

Based on the PRISMA guidelines for network meta-analysis, the study further explored the relative effects of different mindfulness interventions. A network evidence plot was used to visualize the relationships between interventions, where the size of the nodes represented the sample size, and the thickness of the connecting lines indicated the number of studies. The inconsistency factor and its 95% confidence interval (CI) were used to assess the consistency of each closed loop, and a consistency model was employed when $p > 0.05$ (*Shim et al., 2017*). The relative ranking of intervention effects was determined using the

surface under the cumulative ranking (SUCRA) curve, with SUCRA values ranging from 0 to 1, where higher values indicated better effectiveness (*Salanti, Ades & Ioannidis, 2011*). A funnel plot was employed to detect publication bias or small-study effects (*Chen et al., 2024*).

## RESULTS

### Literature selection

A total of 2,226 studies were identified through database searches: PubMed ($n = 383$), Cochrane ($n = 1,073$), Embase ($n = 261$), Web of Science core collection ($n = 475$), CNKI ($n = 33$), and other sources ($n = 1$). Following the PRISMA guidelines, duplicate studies were first removed ($n = 731$). Screening of titles and abstracts excluded the following: screening of titles and abstracts led to the exclusion of articles related to meta-methodology or unrelated to mindfulness ($n = 12$), non-RCT studies ($n = 7$), reviews ($n = 32$), irrelevant topics ($n = 1,300$), and clinical trial registrations ($n = 5$). Subsequently, full-text analyses were conducted, leading to the exclusion of studies due to: outcome measures not meeting inclusion criteria ($n = 36$), interventions not meeting inclusion criteria ($n = 36$), studies on mindfulness-based cognitive therapy ($n = 9$), study populations not meeting criteria ($n = 38$), and unavailable full texts ($n = 1$). Ultimately, 19 studies were included, of which 12 included both depression and anxiety outcomes, zero included depression outcomes only, and seven included anxiety outcomes only (as shown in Fig. 1).

### Basic features of literature inclusion

The basic characteristics of the included studies are summarized in Table 1. These studies were conducted in the USA, China, Portugal, and Spain, involving a total of 1,697 participants. Participants were university students without diagnosed psychiatric disorders, with no psychotropic medication use reported. The interventions included various types of mindfulness-based or physical exercises, such as yoga, tai chi, and qigong, with session durations ranging from 60 to 400 min per week, delivered under supervised conditions. Control groups typically involved no exercise or routine daily activities. Outcomes were measured using validated instruments, including the State-Trait Anxiety Inventory (STAI), Symptom Checklist-90 (SCL-90), Depression Anxiety Stress Scales-21 (DASS-21), and Generalized Anxiety Disorder Scale (GAD-7). Overall, the included studies demonstrated a diversity of interventions and consistent evidence supporting their positive effects on mental health outcomes among university students.

### Risk of bias assessment results for included studies

The risk of bias assessment results of the 19 included studies are shown in Figs. S1 and S2. Among these studies, 18 explicitly described the method of random sequence generation, and all were rated as low risk for selection bias. However, allocation concealment was clearly reported in only 15 studies, leaving four studies rated as unclear risk in this domain. Regarding blinding, 12 studies blinded participants, and seven studies implemented blinding of both participants and outcome evaluators. Due to practical limitations, some studies failed to blind the evaluators, which led to high detection bias risk in three studies.

**Table 1  Characteristics of the included studies.**

| Study | Country | Characteristics of subject | | | | Interventions information | | | | | Outcome (include) | |
|---|---|---|---|---|---|---|---|---|---|---|---|---|
| | | No. (F/M) | Age (M[SD]) | U.M. | BMI(M[SD]) | T.E. | Duration and intensity | V | P.F. | Sup. | Outcomes | Instrument |
| *Albracht-Schulte & Robert-McComb (2018)* | USA | 20F | 20.2 (1.97) | no | 23.2 (3.12) | Yoga Fit first | day 1:30 min (moderate Yoga Fit), day2:30 min (moderate Yoga Fit); moderate | 30 | 1×2 | supervised | Anxiety; | STAI-Y |
| | | 20F | 20.2 (1.97) | no | 23.2 (3.12) | quiet rest first | day1: 30 min (seated and quiet rest), day 2: 30 min (seated and quiet rest); moderate | 30 | 1×2 | supervised | | |
| *Alzahrani et al. (2023)* | Qatar | 39(24/15) | 22.2 (1.70) | no | / | MBSR (Yoga) | 150 min/week, he third class is yoga; moderate | 150 | 8×150 | supervised | Stress; anxiety; | PSS;PHQ-9; GAD-7 |
| | | 45(30/15) | 22.4 (1.70) | no | / | waitlist | no exercise | / | / | / | | |
| *Brandão et al. (2024)* | Portugal | 28(24/4) | 21.5 (5.70) | no | / | Yoga | 60 min/time, on line; | 60 | 6×60 | supervised | depression; anxiety; stress; | DASS |
| | | 34(32/2) | 21.4 (4.64) | no | / | Auto-traini | 60 min/time, on line; | 60 | 6×60 | non | | |
| | | 44(40/4) | 20.5 (3.75) | no | / | no exercise | no exercise | / | / | / | | |
| *Caldwell et al. (2016)* | USA | 28(20/8) | 21.2 (3.00) | no | / | TCC | 10 weeks of TCC meeting 2 times per week, 60 min/-time | 120 | 10×120 | supervised | State anxiety | STAI-Y |
| | | 28(19/9) | 20.2 (1.20) | no | / | Enhanced TCC | 10 weeks of TCC with a DVD of the curriculum, 60 min/time | 120 | 10 ×120 | supervised | | |
| | | 19(14/5) | 22.4(5.50) | no | / | Control | no exercise | / | / | / | | |
| *Castellote-Caballero et al. (2024)* | Spain | 65(32/33) | 20.3(1.78) | no | 24.1(2.60) | Yoga | 60 min/time, 2 times/week; | 120 | 12×120 | supervised | State, Trait anxiety | STAI-Y |
| | | 64(34/30) | 20.3(1.76) | no | 23.3(2.53) | Control | no exercise | / | / | / | | |
| *Chen et al. (2009)* | China | 42 | 20.3(1.50) | no | / | yoga | 120 min/time, | 120 | 16 × 120 | / | depression; anxiety; | SCL-90 |
| | | 49 | 20.3(1.50) | no | / | Control | no exercise | / | / | / | | |
| *Erdoğan Yüce & Muz (2020)* | Turkey | 44(39/5) | 20.0(1.25) | no | / | Yoga | 60 min/week; | 60 | 4 × 60 | supervised | State and trait anxiety | STAI |
| | | 45(41/4) | 19.8(1.04) | no | // | Control | no exercise | / | / | / | | |
| *Falsafi (2016)* | USA | 21 | 22.1 | no | / | mindful | 20 min/day, 75 min/week; | 75 | 8 × 75 | supervised | depression; anxiety; stress; | Beck Hamilton |
| | | 23 | 22.1 | no | / | yoga | 20 min/day,75 min/week; | 75 | 8×75 | supervised | | |
| | | 23 | 22.1 | no | / | Control | no exercise | / | / | / | | |
| *Fu et al. (2019)* | China | 80F | 19.8(1.04) | no | 19.5( 3.05) | Yoga | 90 min/time, 2 time/week; | 180 | 17× 180 | supervised | depression; anxiety; | SCL90 |
| | | 80F | 20.3(1.01) | no | 19.6(2.57) | control | no exercise | / | / | / | | |
| *Hua & Sun (2021)* | China | 67(66/1) | 20.3(0.84) | no | / | Yoga | >60 min/week; | >60 | 12×120 | supervised | depression; anxiety; stress; | DASS |
| | | 60(39/21) | 20.4(0.80) | no | / | TCC | >60 min/week; | >60 | 12 × 120 | supervised | | |
| | | 63(62/1) | 20.2(0.99) | no | / | exercise | >60 min/week; | >60 | 12 × 120 | supervised | | |
| | | 20(10/10) | 21.2(1.04) | no | / | Control | no exercise | / | / | / | | |
| *Jiao, Ji & Chen (2021)* | China | 40F | 20.0(1.3) | No | / | Wuqinxi | 80 min/time; 5 times/week; | 400 | 16 ×400 | supervised | depression; anxiety; | SCL-90 |
| | | 40F | 20.0(1.3) | no | / | Control | no exercise | / | / | / | | |

**Table 1** (*continued*)

| Study | Country | Characteristics of subject | | | | Interventions information | | | | | Outcome (include) | |
|---|---|---|---|---|---|---|---|---|---|---|---|---|
| | | No. (F/M) | Age (M[SD]) | U.M. | BMI(M[SD]) | T.E. | Duration and intensity | V | P.F. | Sup. | Outcomes | Instrument |
| *Li et al. (2023)* | China | 195(89/106) | 24.0(4) | no | / | Baduanjin | 45 min/time; 5 times/week; | 225 | 11 × 225 | supervised | anxiety | CAS |
| | | 192(101/91) | 23.0(3) | no | / | control | no exercise | / | / | / | | |
| *Li, Yang & Zhang (2023)* | China | 24F | 18–26 | no | / | wuqinxi | 30 min/time; 5 times/week; | 150 | 12 × 150 | supervised | Anxiety; | SCL-90 |
| | | 24F | 18–26 | no | / | Wuqinxi +mental | PM 22:00, 35 min/time; 5 times/week ; mental com-munication | 175 | 12 × 150 | supervised | | |
| | | 24F | 18–26 | no | / | metal | 30 min/time; 1 time/week; | / | / | / | | |
| *Papp et al. (2019)* | Sweden | 21 (18/3) | 20–37 | no | 23.73( 8.80) | Yoga | 60 min/week; high inten-sity; | 60 | 6 × 60 | supervised | depression; anxiety; | HADS |
| | | 23(20/3) | 20–39 | no | 22.25(7.79) | Control | no exercise | | | | | |
| *Sun et al. (2024)* | China | 18(11/7) | 21.0(5.1) | no | 21.23(5.82) | Qigong | 60 min/time, 5 times/week; | 300 | 12 × 300 | supervised | Physical, Mental anxiety | HAS |
| | | 19(8/11) | 20.2(3.5) | no | 19.76(2.06) | Control | No exercise | / | / | / | | |
| *Xiao et al. (2021)* | China | 31(8/23) | 18.9(0.89) | no | / | basketball | 90 min/time; 3 times/week; moderate | 270 | 12 × 270 | supervised | anxiety | SAS |
| | | 31(7/24) | 19.2(1.02) | no | / | baduanjin | 90 min/time; 3 times/week; moderate | 270 | 12 × 270 | supervised | | |
| | | 34(10/24) | 19.7(1.77) | no | / | Control | no exercise | | | | | |
| *Zhang et al. (2023)* | China | 9(7/2) | 24.2(4.07) | no | 21.92(4.38) | Tai Chi | 60 min/time, 5 times/week; | 300 | 8 × 300 | supervised | depression; anxiety; | SAS; SDS |
| | | 9(6/3) | 22.5(5.95) | no | 20.96(4.65) | Control | no exercise | / | / | / | | |
| *Zhang & Jiang (2023)* | China | 39F | 19.2(0.98) | no | 20.71(3.43) | baduanjin | 60 min/time, 3 times/week; | 180 | 12 × 180 | supervised | depression; anxiety; | SCL-90 |
| | | 39F | 19.16(1.05) | no | 21.03(3.87) | Control | normal study life | / | / | / | | |
| *Zheng et al. (2018)* | China | 17(11/6) | 35.4(2.1) | no | / | TCC | 120 min/week, 6 week; 6 week, home practice | 120 | 12 × 120 | supervised | depression; anxiety; | STAI |
| | | 17(14/3) | 32.0(1.8) | no | / | exercise | 300 min/week; | 300 | 12 × 300 | supervised | | |
| | | 16(14/2) | 34.6(2.3) | no | / | Control | normal study life | / | / | / | | |

**Notes.**

*S, study; C, Country; F/M, Female/Male; M[SD], Mean[SD]; T.E, Type of exercise; UM, Use of psychotropic or medications; V, Volume (min/week); P.F., Period and frequency (week × [times/week]); SUP, supervised or non-supervised; AE, aerobic exercise; STAI-Y2, Trait Anxiety Inventory; F, female; MBSR, Mindfulness-Based Stress Reduction; PHQ-9, Patient health questionnaire 9; MAAS, Mind-ful attention awareness scale; GAD-7, Generalized anxiety disorder scale 7; PSS, perceived stress scale; TCC, tai chi Chuan; PANAS, Positive and Negative Affect Schedule; SAS, Self-rating Anxiety Scale; CAS, The coronavirus anxiety scale; HADS, The Hospital Anxiety and Depression Scale; HAS, Hamilton Anxiety Scale.
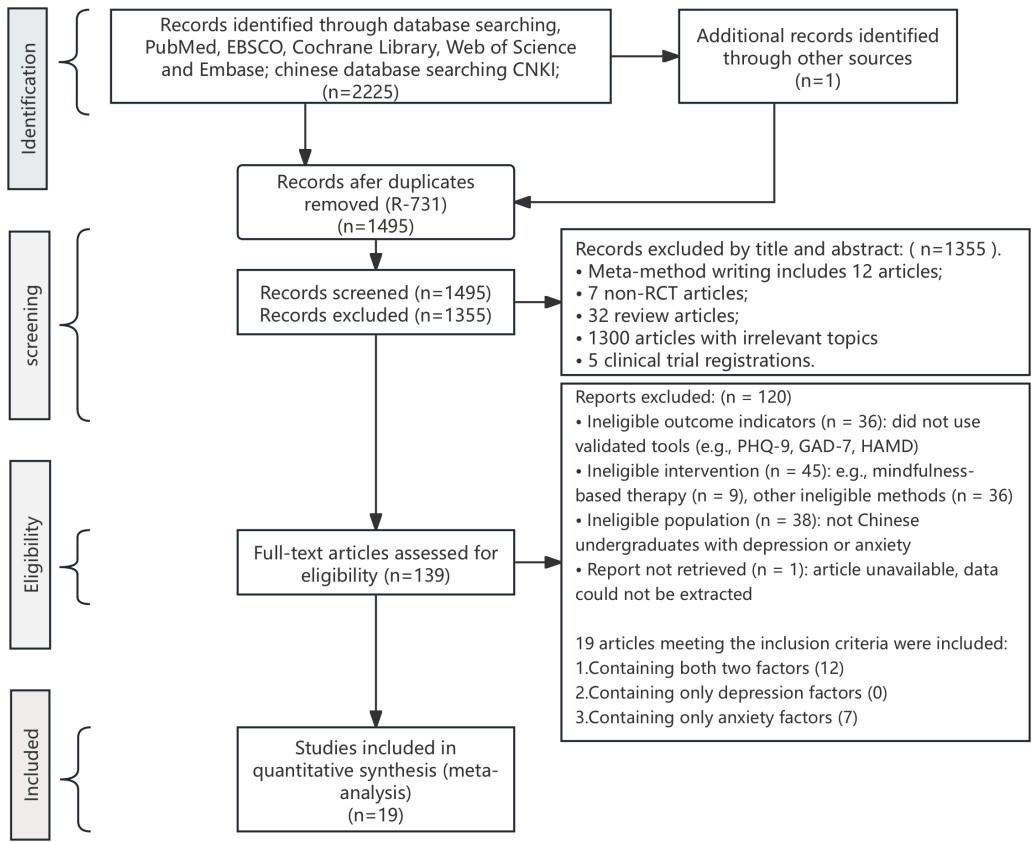

**Figure 1  Flow diagram of systematic literature search.**

For data completeness, all 19 studies provided detailed descriptions of dropout rates and reasons for attrition. However, one study had dropout rates exceeding 20% and did not employ intention-to-treat analysis, which were consequently rated as high risk for attrition bias. In the domain of other biases, one study had a small sample size (<10) and was rated as high risk due to potential small-sample bias. Additionally, no significant conflicts of interest were reported in any study. Overall, 16 studies were rated as high quality, three as moderate quality, and one as low quality based on the combined evaluation of all risk domains. In *Chen et al. (2009)*, the study was rated as high risk due to missing outcome data and the absence of baseline (pre-test) measurements. As a result, it was excluded from the quantitative synthesis and not included in the meta-analysis.

## Mindful movement's effect on anxiety
### Pooled effect size analysis of anxiety-related outcomes

The forest plot (Fig. 2) of anxiety-related outcomes indicates that mindful movement interventions significantly improve anxiety levels among university students. The pooled effect size was SMD = −0.76 (95% CI [−1.07 to −0.45]), $p < 0.01$, suggesting that the effect size (−0.76) represents a statistically significant mean difference. Moreover, the highly significant $p$-value (<0.0001) indicates that the mindfulness-based intervention

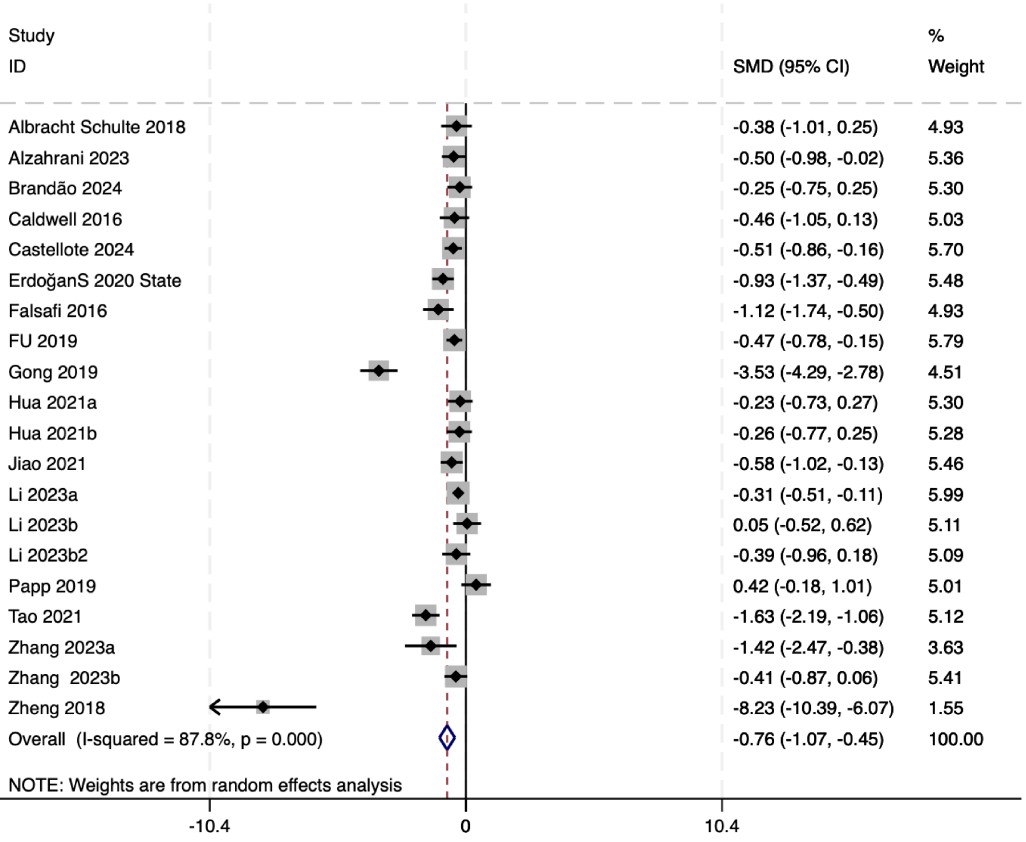

**Figure 2  The forest plot of anxiety-related outcomes.** Notes. *Albracht-Schulte & Robert-McComb (2018)*, *Alzahrani et al. (2023)*, *Brandão et al. (2024)*, *Caldwell et al. (2016)*, *Castellote-Caballero et al. (2024)*, *Erdoğan Yüce & Muz (2020)*, *Falsafi (2016)*, *Fu et al. (2019)*, *Gong et al. (2019)*, *Hua & Sun (2021)*, *Jiao, Ji & Chen (2021)*, *Li et al. (2023)*, *Li, Yang & Zhang (2023)*, *Papp et al. (2019)*, *Xiao et al. (2021)*, *Zhang & Jiang (2023)*, *Zheng et al. (2018)*.

group showed a statistically significant improvement compared to the control group. The heterogeneity test revealed $I^2 = 87.8\%$, indicating a moderate degree of heterogeneity among the studies.

### Sensitivity analysis of anxiety-related outcomes

After the sensitivity analysis (as shown in Fig. 3), the overall heterogeneity was significantly reduced from $I^2 = 87.8\%$ to $I^2 = 61.8\%$, indicating that the excluded studies contributed substantially to the observed heterogeneity. The pooled effect size decreased from SMD = $-0.76$ (95% CI [$-1.07$ to $-0.45$]) to SMD = $-0.49$ (95% CI [$-0.67$ to $-0.31$]). Sensitivity analyses indicated that studies with longer intervention durations tended to report larger effect sizes for anxiety outcomes, suggesting a potential dose–response relationship. While the effect size was slightly attenuated, it remained statistically significant, suggesting the effectiveness of mindfulness-based interventions in alleviating anxiety symptoms among university students. These results demonstrate the robustness of the overall findings and
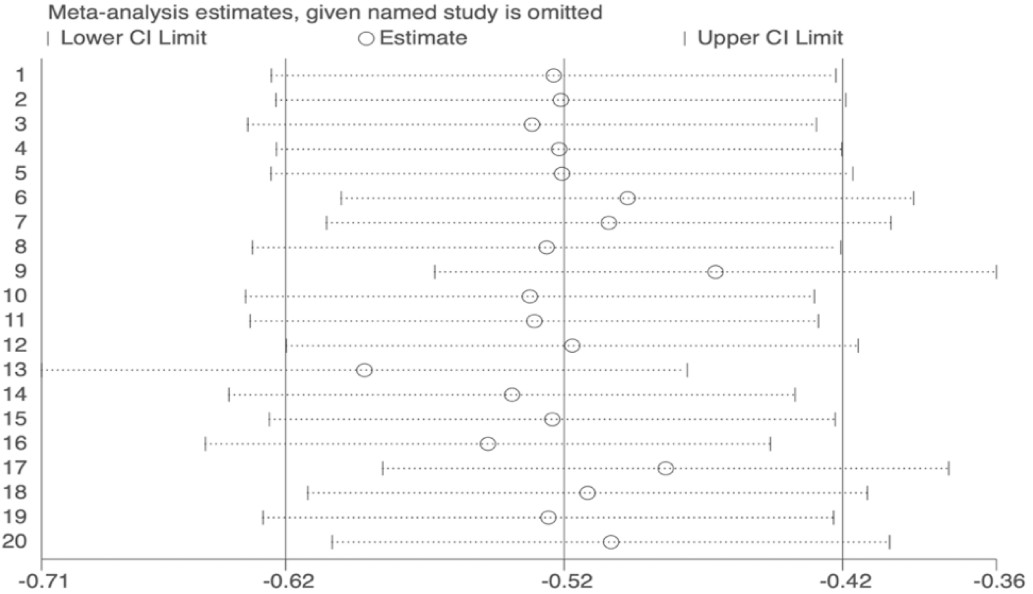

**Figure 3** The sensitivity analysis of anxiety-related outcome.

highlight the influence of specific studies on heterogeneity, suggesting that certain study characteristics may substantially impact variability. In sensitivity analysis, we systematically excluded each study one at a time (leave-one-out analysis) to assess the impact on overall heterogeneity and effect size.

*Subgroup analysis of anxiety factor*

The subgroup analysis results (as shown in Fig. 4) indicate that mindful movement interventions (yoga, tai chi, qigong) significantly reduce anxiety levels among university students, with varying effect sizes and heterogeneity. Yoga demonstrated a moderate pooled effect size (SMD = −0.47, 95% CI [−0.62 to −0.32]) with moderate heterogeneity ($I^2 = 57.0\%$, $P = 0.02$), suggesting that factors such as intervention duration and participant characteristics may contribute to the variability. Tai chi showed a slightly smaller but still significant effect size (SMD = −0.47, 95% CI [−0.84 to −0.11]) with lower heterogeneity ($I^2 = 47.9\%$, $p = 0.15$), indicating moderate reliability. Qigong exhibited a stable and consistent effect size (SMD = −0.43, 95% CI [−0.58 to −0.27]) with relatively high heterogeneity ($I^2 = 77.2\%$, $p = 0.00$), suggesting potential variability due to study design or intervention protocols.

Overall, the pooled effect size for mindful movement interventions was SMD = −0.45 (95% CI [−0.55 to −0.35]), with moderate overall heterogeneity ($I^2 = 61.8\%$, $p < 0.0001$). The test for subgroup differences indicated no statistically significant differences between intervention types ($p = 0.92$), suggesting that yoga, tai chi, and qigong exhibit comparable efficacy in reducing anxiety levels among university students.

*Meta-regression analysis of anxiety factor*

The results (Table 2) of the meta-regression conducted on 18 studies indicated that the covariate "type" had no significant effect on the effect size (exp(b) = 0.96, $p = 0.75$),

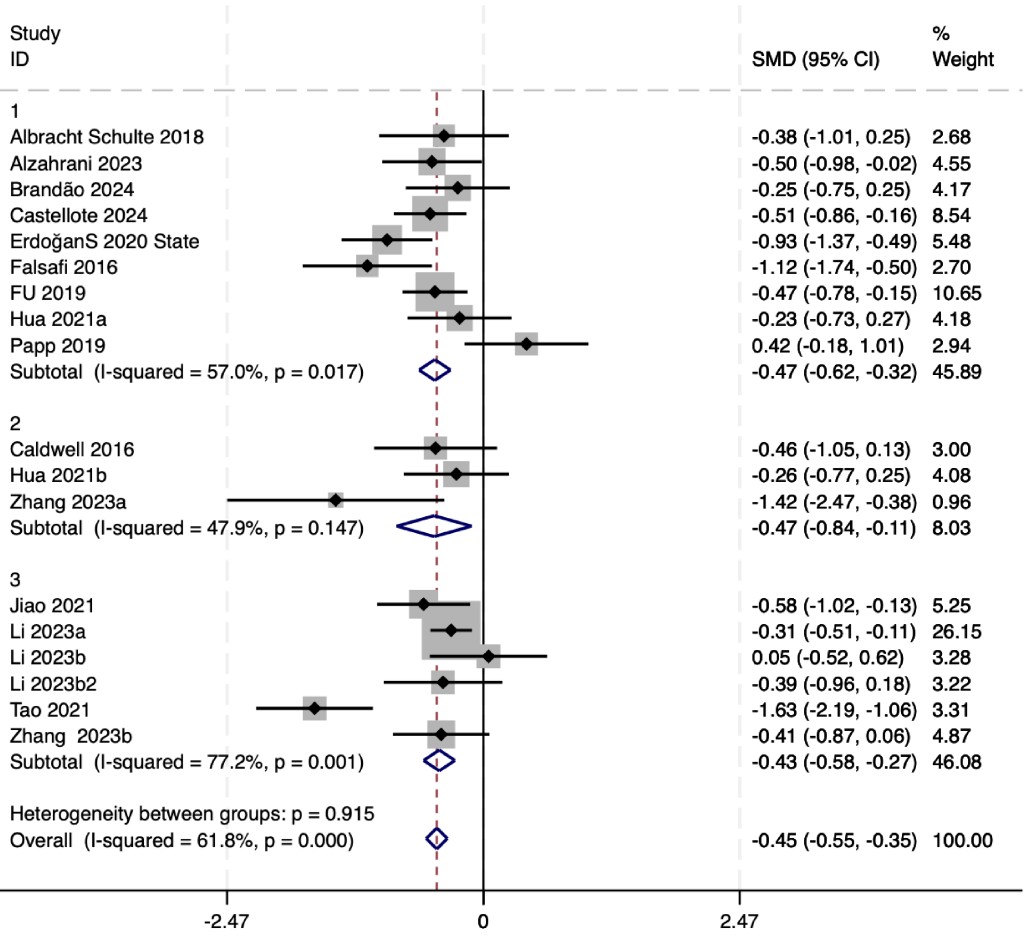

**Figure 4  Subgroup analysis of anxiety.** Notes. *Albracht-Schulte & Robert-McComb (2018)*, *Alzahrani et al. (2023)*, *Brandão et al. (2024)*, *Caldwell et al. (2016)*, *Castellote-Caballero et al. (2024)*, *Erdoğan Yüce & Muz (2020)*, *Falsafi (2016)*, *Fu et al. (2019)*, *Gong et al. (2019)*, *Hua & Sun (2021)*, *Jiao, Ji & Chen (2021)*, *Li et al. (2023)*, *Li, Yang & Zhang (2023)*, *Papp et al. (2019)*, *Xiao et al. (2021)*, *Zhang & Jiang (2023)*, *Zheng et al. (2018)*.

suggesting a limited influence on the overall effect size. The high residual heterogeneity (I-squared_res = 63.92%) and the negative adjusted R-squared indicate that including "type" in the regression model not only failed to explain between-study differences but also may have introduced additional unexplained heterogeneity. Moreover, neither the intercept ($p = 0.101$) nor the covariate reached statistical significance, suggesting that "type" had limited explanatory power for effect size differences across studies in this model.

This result was further confirmed by the regression forest plot, where the SMDs of the three subgroups (1, 2, and 3) were −0.47, −0.47, and −0.43, respectively, with a total SMD of −0.45 (see attached map 2). These results demonstrate no significant differences in the overall effect of different types of mindful movement interventions (*e.g.*, yoga, tai chi, and qigong) on anxiety among university students. Thus, mindful movement can be treated

**Table 2  Meta-regression results of anxiety.**

|  |  | ES | exp(b) | Std. err. | t | P >|t| | [95% CI] | |
|---|---|---|---|---|---|---|---|---|
| With Knapp-Hartung modification | type | .9619132 | .1136414 | −0.33 | 0.747 | .7488034 | 1.235674 |
|  | _cons | .6574694 | .1595963 | −1.73 | 0.103 | .3929983 | 1.099918 |

**Notes.**

Meta-regression (Number of obs = 18);
REML estimate of between-study variance (tau2 = .1206);
Residual variation due to heterogeneity (I-squared_res = 63.92%);
Proportion of between-study variance explained (Adj R-squared = −14.35%).

**Table 3  Begg and Egger bias test of anxiety.**

| Type |  | P | All | Yoga | TCC | Qigong |
|---|---|---|---|---|---|---|
| Begg's Test |  | Pr >|z| | 0.850 | 0.404 | 0.117 | 0.851 |
|  | Cont. corrected | Pr >|z| | 0.880 | 0.466 | 0.296 | 1 |
| Egger's test | Slope | P>|t| | 0.334 | 0.220 | 0.026 | 0.749 |
|  | Bias | P>|t| | 0.327 | 0.658 | 0.016 | 0.445 |

as a proxy for these interventions, supporting a unified analysis of their overall effect on anxiety reduction.

## Publication bias test of anxiety factor

Based on these publication bias tests (Table 3), most categories (all, yoga, and qigong) do not show statistically significant bias. Both Begg's and corrected Begg's tests yield $p$-values greater than 0.05, and the Egger's test also shows non-significant slopes and bias terms for these groups, indicating no strong evidence of missing studies or publication bias. However, for the tai chi group, the Egger's test reveals significant results (slope $p = 0.03$, bias $p = 0.02$), suggesting potential publication bias (bias assessment funnel plot; as shown in Fig. S3).

## Mindful movement's effect on depression
### Pooled effect size analysis of depression-related outcomes

The meta-analysis results for mindfulness-based interventions on depression (as shown in Fig. 5) among university students indicate a significant overall effect, with a pooled standardized mean difference (SMD) of −0.73 (95% CI) [−1.08 to −0.38]), favoring the intervention group ($p < 0.00$). However, substantial heterogeneity was observed ($I^2 = 81.9\%$), suggesting considerable variability across studies. This heterogeneity may stem from differences in intervention types, durations, participant characteristics, or study designs. These results suggest that mindfulness-based interventions are effective in alleviating depressive symptoms among university students. Still, the high level of heterogeneity highlights the necessity of further investigations to identify and address the sources of variability, potentially through more standardized intervention protocols or subgroup analyses.

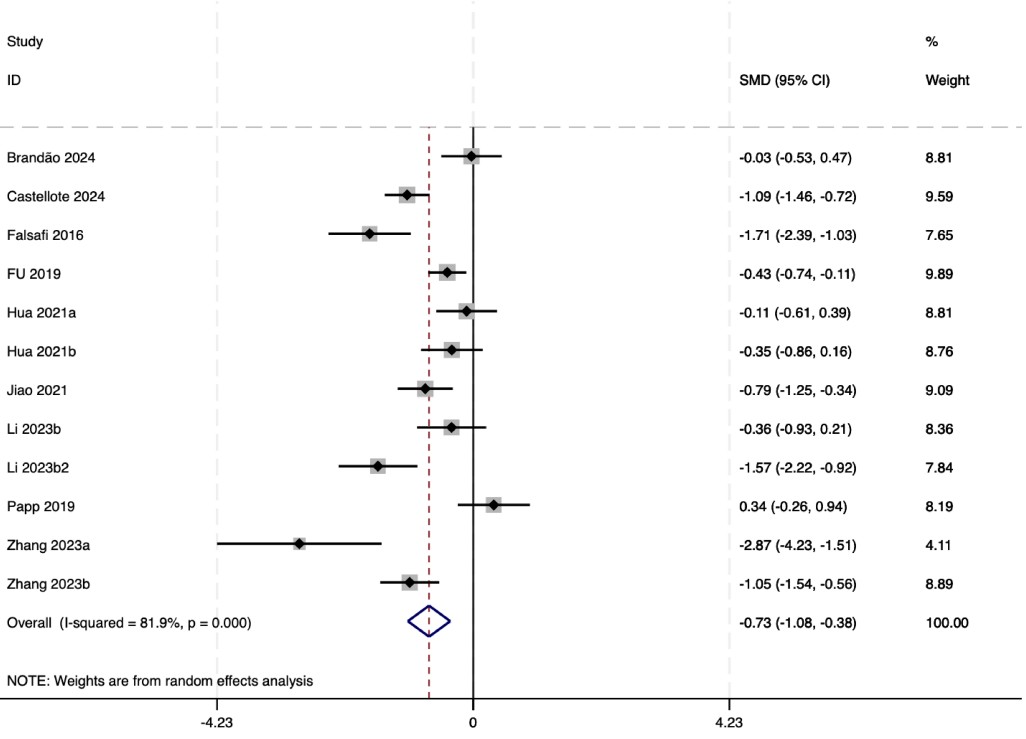

**Figure 5  The forest plot of depression-related outcomes.** Notes. *Brandão et al. (2024)*, *Caldwell et al. (2016)*, *Castellote-Caballero et al. (2024)*, *Falsafi (2016)*, *Fu et al. (2019)*, *Hua & Sun (2021)*, *Jiao, Ji & Chen (2021)*, *Li et al. (2023)*, *Li, Yang & Zhang (2023)*, *Papp et al. (2019)*, *Zhang & Jiang (2023)*.

### Sensitivity analysis of depression-related outcomes

The sensitivity analysis results, as shown in Fig. 6, demonstrate the robustness of the meta-analysis results for depression-related outcomes. After systematically omitting individual studies, the pooled standardized mean difference (SMD) remained stable, with values ranging between −0.78 and −0.50. This consistent range indicates that no single study disproportionately influenced the overall effect size. Additionally, the confidence intervals remained narrow across all iterations, further supporting the reliability of the results. These results suggest that the observed significant reduction in depressive symptoms attributed to mindful movement interventions is not driven by any specific study and highlights the robustness and generalizability of the meta-analysis conclusions.

### Subgroup analysis of depression factor

The subgroup analysis results (as shown in Fig. 7) presented in the forest plot reveal the differential impact of mindful movement interventions on depressive symptoms across distinct subgroups. The overall pooled standardized mean difference (SMD) remains significant at −0.64 (95% CI [−0.78 to −0.50]), indicating a strong intervention effect.

Subgroup one (largest studies) shows a pooled SMD of −0.58 (95% CI [−0.75 to −0.41]), with moderate heterogeneity ($I^2 = 84.8\%$, $p = 0.0001$). This suggests variability among studies in this subgroup, potentially due to differences in intervention protocols

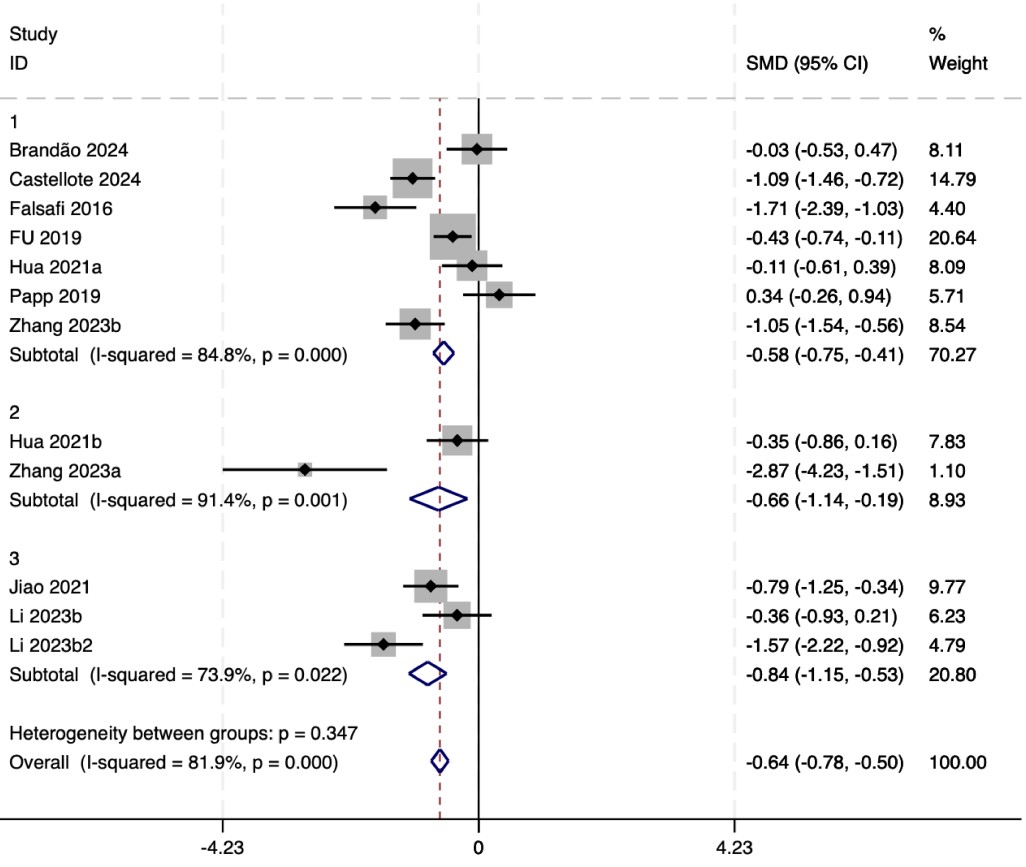

**Figure 6** **Subgroup analysis of depression.** Notes. *Brandão et al. (2024)*, *Castellote-Caballero et al. (2024)*, *Falsafi (2016)*, *Fu et al. (2019)*, *Hua & Sun (2021)*, *Jiao, Ji & Chen (2021)*, *Li et al. (2023)*, *Li, Yang & Zhang (2023)*, *Papp et al. (2019)*, *Zhang & Jiang (2023)*.

or participant characteristics. Subgroup two (smaller studies) shows the effect size here is −0.66 (95% CI [−1.14 to −0.19]), with high heterogeneity ($I^2 = 91.4\%$, $p = 0.001$). The significant heterogeneity could be attributed to limited study sizes or methodological variations, impacting the consistency of results. Subgroup three (specialized studies) reports the largest effect size at −0.84 (95% CI [−1.15 to −0.53]), with lower heterogeneity ($I^2 = 73.9\%$, $p = 0.022$). This may indicate a more focused or tailored intervention effect in this context.

Despite heterogeneity within subgroups, the heterogeneity between groups ($p = 0.35$) is not significant, suggesting that the differences between these subgroups may not be statistically significant. The consistent significant effect sizes across all subgroups highlight the efficacy of mindful movement interventions for reducing depressive symptoms. However, the varying levels of heterogeneity emphasize the need for further exploration into potential modifiers such as intervention duration, participant demographics, or study design to optimize intervention outcomes.
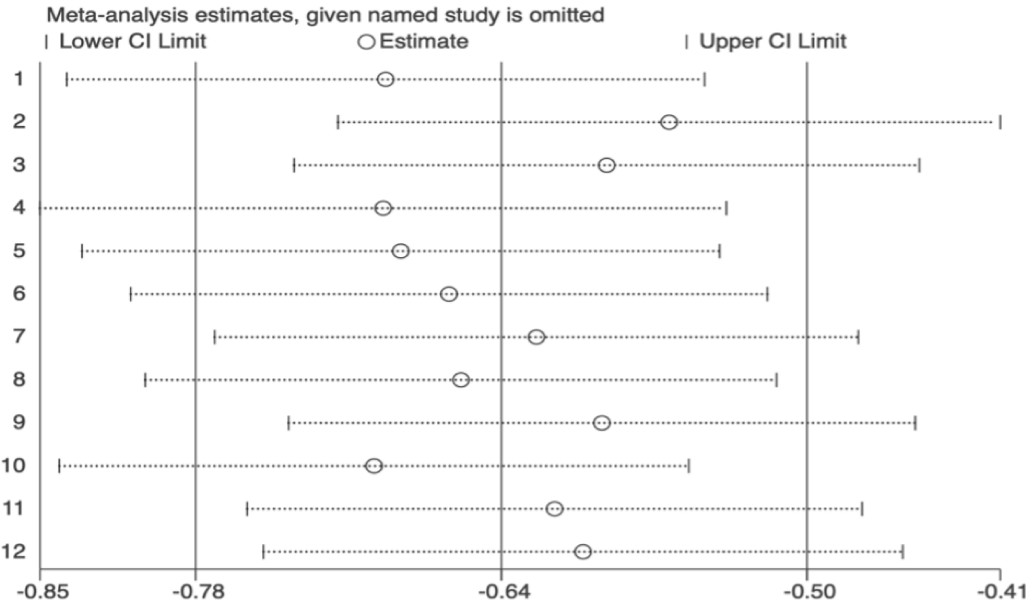

**Figure 7** The sensitivity analysis of depression.

**Table 4** Meta-regression results of depression.

|  | ES | exp(b) | Std. err. | t | P>\|t\| | [95% CI] | |
|---|---|---|---|---|---|---|---|
| With Knapp-Hartung modification | type | .8233249 | .2262242 | −0.71 | 0.495 | .4463606 | 1.518646 |
|  | _cons | .6494626 | .3308544 | −0.85 | 0.417 | .2087358 | 2.020745 |

**Notes.**
*Meta-regression (Number of obs = 12); REML estimate of between-study variance (tau2 = .4802);
Residual variation due to heterogeneity (I-squared_res = 83%);
Proportion of between-study variance explained (Adj R-squared = −8.61%).

### *Meta-regression analysis of depression factor*

The meta-regression analysis, based on 12 observations, evaluated the influence of "type" as a covariate on the effect size (ES). The Knapp-Hartung modification was applied to enhance the robustness of the estimates. The results indicate that "type" is not a statistically significant predictor of the effect size, as demonstrated by its *p*-value ($p = 0.495$) and the wide 95% confidence interval (95% CI [0.45–1.52]), which includes one. This suggests that variations in the type of intervention do not explain the differences in effect size across studies (see Table 4).

The residual variation due to heterogeneity (I-squared_res = 83%) remains high, implying substantial unexplained heterogeneity. Furthermore, the adjusted R-squared value indicates that the inclusion of "type" as a covariate did not improve the explanatory power of the model; instead, it might have slightly increased the unexplained heterogeneity.

### Publication bias test of depression factor

The publication bias test results for all groups, as shown in Table 5, indicate no statistically significant evidence of bias across the majority of categories. Begg's test and the corrected

**Table 5  Begg and Egger bias test of depression.**

| Type | | P | All | Yoga | TCC | Qigong |
|---|---|---|---|---|---|---|
| Begg's Test | | Pr >\|z\| | 0.217 | 0.652 | 0.317 | 0.602 |
| | Cont. corrected | Pr >\|z\| | 0.244 | 0.764 | 1.000 | 1 |
| Egger's test | slope | P>\|t\| | 0.997 | 0.560 | / | 0.880 |
| | bias | P>\|t\| | 0.277 | 0.970 | / | 0.697 |

Begg's test yield $p$-values greater than 0.05 in all groups (*e.g.*, all: Pr > |z| = 0.25, yoga: Pr > |z|= 0.65), suggesting no significant asymmetry in the funnel plots. Similarly, the Egger's test results, where available, also show no significant bias (*e.g.*, all: bias $p$ > |t| = 0.28, yoga: bias $p$ > |t|= 0.97). These results indicate a low likelihood of missing studies or small-study effects for these groups. For the tai chi group, due to the inclusion of only two studies and high variability, Egger's test could not generate valid results. This limitation highlights the reduced reliability of bias assessments for this subgroup, as the small sample size undermines the robustness of the statistical tests (bias assessment funnel plot; as shown in Fig. S4).

Overall, while the analysis suggests no significant publication bias in most groups, the limited data in the tai chi group warrants caution in interpreting these results. Future studies with larger sample sizes in this subgroup would help provide more conclusive evidence regarding the presence or absence of publication bias.

## Certainty of evidence
### Certainty of evidence for anxiety outcomes
The evidence indicates that mindful movement interventions significantly reduce anxiety among university students (SMD = −0.76, 95% CI [−1.07 to −0.45], $p < 0.01$). Risk of bias was minimized as most studies were high-quality RCTs, though some protocol variations were noted. Inconsistency was moderate ($I^2 = 87.8\%$) but reduced to $I^2 = 61.8\%$ after sensitivity analysis. Indirectness was not a concern, as all studies directly targeted anxiety symptoms. Imprecision was low, with narrow confidence intervals, even after adjustment (SMD = −0.49, 95% CI [−0.67 to −0.31]). Publication bias was not significant overall ($p >$ 0.05), except in the tai chi subgroup (Egger's test $p = 0.03$).

### Certainty of evidence for depression outcomes
The evidence indicates that mindful movement interventions significantly reduce depressive symptoms among university students (SMD = −0.73, 95% CI [−1.08 to −0.38], $p < 0.00$). Risk of bias was low for most studies, though smaller studies showed methodological limitations. Inconsistency was substantial ($I^2 = 81.9\%$), but subgroup analysis confirmed significant reductions across all categories. Indirectness was not an issue, as all studies directly addressed depressive symptoms. Imprecision was moderate, with slightly wider confidence intervals, but consistent findings in sensitivity and subgroup analyses supported robustness. Publication bias was not significant ($p >$ 0.05) overall, though the tai chi subgroup lacked sufficient data for reliable assessment.

**Summary**

The evidence suggests that mindful movement interventions are effective in reducing both anxiety and depressive symptoms among university students. The overall certainty of evidence is moderate for anxiety and depression, due to heterogeneity and potential biases.

# DISCUSSION

This meta-analysis demonstrates that mindfulness-based interventions significantly improve anxiety and depressive symptoms among university students (*Alzahrani et al., 2023*; *Falsafi, 2016*). The pooled effect sizes reveal that mindful movement interventions significantly reduce anxiety (SMD = −0.42, 95% CI [−0.52 to −0.31]) and depression (SMD = −0.61, 95% CI [−1.01 to −0.20]), with the overall intervention effect favoring the experimental group over the control group. These results align with previous studies (*Breedvelt et al., 2019*), which have focused on RCTs exploring mindful movement interventions. The results validate that diverse intervention types, such as yoga, tai chi, and qigong, positively influence depressive and anxiety symptoms in university students (*Hua & Sun, 2021*; *Sun et al., 2024*; *Zhang et al., 2023*). The SMD reflects a moderate-to-strong effect of these interventions in alleviating anxiety and depression, highlighting the potential for broader application and dissemination.

Depression and anxiety are key indicators of mental health disorders (*Fodor et al., 2018*) and represent common psychological challenges faced by university students. Yoga-based physical activities effectively reduce perceived stress and state anxiety and are suggested as an effective approach for alleviating stress and anxiety among young adults (*Erdoğan Yüce & Muz, 2020*; *Fu et al., 2019*). Long-term engagement in mindfulness-based physical practices significantly reduces anxiety, depression, and stress while enhancing mindfulness levels among university students (*Hua & Sun, 2021*). Moreover, qigong, as a primary form of mindfulness-based exercise, has been shown to improve both physical fitness and mental health in female university students, serving as an effective and practical fitness intervention (*Jiao, Ji & Chen, 2021*).

The subgroup analysis revealed no statistically significant differences in intervention effects among the different types of mindfulness-based exercises (*Du et al., 2022*). Various forms of mindfulness practices, such as yoga, tai chi, and qigong, demonstrated effectiveness in alleviating anxiety and depressive symptoms. However, substantial variability in intervention effects and heterogeneity was observed. For instance, qigong exhibited the most pronounced effect on anxiety, with findings consistent with existing literature (*Li, Yang & Zhang, 2023*; *Zhang & Jiang, 2023*). This subgroup showed the highest standardized mean difference (SMD), along with a notable reduction in heterogeneity. These results suggest potential subgroup-specific mechanisms underlying the effects of different mindfulness exercises on anxiety and depression in university students (*Xiao et al., 2021*; *Zheng et al., 2018*).

Moreover, sensitivity analysis revealed a significant reduction in heterogeneity for anxiety-related studies after stratification by intervention type ($I^2$ values of 56%, 41%, and

0% for the three subgroups, respectively), indicating that differences in intervention types contributed to the heterogeneity in anxiety outcomes. In contrast, no significant changes were observed in heterogeneity for depressive outcomes, suggesting that depression-related heterogeneity is less influenced by intervention type. Nonetheless, intervention type is unlikely to be the sole determinant of heterogeneity. Other potential sources of variability include differences in intervention duration, frequency, baseline participant characteristics (*e.g.*, gender, age), and study design (*e.g.*, randomization methods, sample size). Consistent with the results described in the sensitivity analyses, studies with longer intervention durations appeared to yield larger effect sizes, suggesting a potential dose–response relationship. Similarly, gender differences in anxiety and depression prevalence may have influenced intervention outcomes, as most studies recruited mixed-gender samples without stratifying results by gender.

The heterogeneity analysis revealed moderate to high heterogeneity ($I^2 = 70\%$–$87\%$), indicating notable variations across the included studies. The meta-regression analysis showed that intervention type did not significantly account for the observed heterogeneity ($p = 0.49$). This non-significance may be attributed to limited statistical power, given the relatively small number of studies included in the meta-regression ($n = 18$). Additionally, intervention type may not fully capture the nuanced differences across studies, such as variations in intervention intensity, duration, and participant baseline characteristics. The high residual heterogeneity ($I^2 = 63.92\%$) highlights the need for future meta-analyses to incorporate additional covariates, such as intervention frequency, baseline psychological health, and cultural context, to better explain between-study differences.

Publication bias was not observed in most subgroups based on funnel plot results, indicating a low overall risk of bias. However, the tai chi subgroup might present potential bias due to the limited number of studies ($n = 2$), and Egger's test could not be performed. These results suggest that the conclusions of this meta-analysis are generally robust. However, considering the presence of moderate-to-high heterogeneity, potential risk of publication bias in specific subgroups, and other methodological limitations, the results should be interpreted with caution, especially regarding individual intervention types.

This study has several limitations. First, the number of included studies was relatively small, particularly for certain subgroups, which may limit the generalizability of the results. Second, substantial differences in intervention duration, sample characteristics, and study designs might exacerbate heterogeneity and influence the consistency of intervention effects. Third, incomplete data reporting. According to the Cochrane Risk of Bias tool, the absence of explicit reporting on randomization methods was rated as high risk, which consequently contributed to the moderate-to-high overall risk of bias across the included studies. Future research should aim to address these limitations by including more high-quality, diverse studies with standardized intervention protocols and comprehensive reporting.

The results of this meta-analysis offer meaningful practical implications for mental health promotion among university students. Given the moderate-to-strong effects of mindful movement interventions such as yoga, tai chi, and qigong, universities and health service providers should consider incorporating these practices into student wellness programs. These interventions are low-cost, culturally adaptable, and can be implemented

in both group and individual formats, making them accessible and sustainable in campus environments. Furthermore, targeted interventions for students experiencing elevated stress or anxiety symptoms may yield particularly beneficial outcomes. Policymakers and educational institutions should recognize the value of these interventions in addressing mental health concerns within the student population and consider integrating them into preventive and supportive care strategies.

## CONCLUSIONS

This study demonstrates that mindful movement interventions significantly improve anxiety and depression symptoms among university students, with some subgroup differences observed in the effects on anxiety factors across different types of mindfulness exercises. Despite the presence of certain heterogeneities and limitations in the included studies, the results provide robust evidence to support the development of mindfulness-based strategies for mental health interventions. Future research should focus on conducting multi-center, high-quality randomized controlled trials to enhance the generalizability and reliability of the results.

## ACKNOWLEDGEMENTS

The authors would like to thank the Department of Sports Training at Universiti Putra Malaysia for their support.

### Funding

This work was supported by the 2019 Higher Education Research Project (Grant No. SHE1906), the 2023 Ideological and Political Demonstration Course Initiative (Grant No. KCSZSFK2322), and the Undergraduate Teaching Case Repository Construction Project of Sanming University. The funders had no role in study design, data collection and analysis, decision to publish, or preparation of the manuscript.

### Grant Disclosures

The following grant information was disclosed by the authors:
2019 Higher Education Research Project: SHE1906.
2023 Ideological and Political Demonstration Course Initiative: KCSZSFK2322.
Undergraduate Teaching Case Repository Construction Project of Sanming University.

### Competing Interests

The authors declare there are no competing interests.

### Author Contributions

- Xinjian Xu conceived and designed the experiments, performed the experiments, analyzed the data, prepared figures and/or tables, authored or reviewed drafts of the article, and approved the final draft.

- Borhannudin Bin Abdullah conceived and designed the experiments, performed the experiments, authored or reviewed drafts of the article, and approved the final draft.
- Shamsulariffin Bin Samsudin conceived and designed the experiments, analyzed the data, authored or reviewed drafts of the article, and approved the final draft.
- Yongneng Tan conceived and designed the experiments, performed the experiments, analyzed the data, prepared figures and/or tables, authored or reviewed drafts of the article, and approved the final draft.

## Data Availability

Because this study is a meta-analysis, all data have been listed in the article.

## Supplemental Information

Supplemental information for this article can be found online at http://dx.doi.org/10.7717/peerj.19640#supplemental-information.

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
