# Peer review of "Integrating mindfulness and physical activity: a meta-analysis of mindful movement interventions for symptoms of anxiety and depression among university students"

_PeerJ, doi:10.7717/peerj.19640_

## Round 0.1 · original submission · Major Revisions

Thank you for your submission. The reviewers have identified a number of concerns that must be addressed, including justification of the review and reporting.

**Language Note:** The review process has identified that the English language must be improved. PeerJ can provide language editing services - please contact us at [email protected] for pricing (be sure to provide your manuscript number and title). Alternatively, you should make your own arrangements to improve the language quality and provide details in your response letter. – PeerJ Staff

·

Basic reporting

Dear authors,
Thank you for the opportunity to read your meta-analysis manuscript on mindful movement interventions. Overall, I think the piece is within the remit of the journal and can contribute to the literature base, after edits for clarity, further demonstration of rationale, and a revised and improved discussion section.

Basic Reporting
1. Language quality needs improvement. You mostly communicate effectively, but there are occasional grammatical and meaning errors, and the need for more precise language. They include:
a. Line 85: What do you mean by mindfulness-base? Substitute for mindful awareness or mindfulness practice, if either are what you mean.
b. L128: Instead of blank control you probably mean waitlist or treatment as usual control?
c. L136: RCTs) that are available in either English or Chinese
d. L202 (and elsewhere): primarily is not appropriate here because it implies non-students in your samples, delete it. You use the word in other parts of the manuscript too, where more precise words would be more appropriate.
e. L236-7: You repeat a sentence on heterogeneity. Revise.
f. L244 & L297: Phrase confirm the effectiveness is not a suitable phrase – the included studies can only fail to reject the null hypothesis; instead, use these findings indicate or suggest
g. L310 (and elsewhere): You refer to mindful movement interventions as mindfulness based interventions, which is not accurate enough (as it includes all sorts of MBIs without any movement. I suggest you revise and say mindful movement interventions throughout the manuscript.
h. L331: may not be statistically meaningful makes no sense – findings are either statistically significant or not, or clinically meaningful or not, but there is not such thing as statistically meaningful. Revise.
i. L448: You repeat sentence on publication bias. Revise.
2. The abstract would benefit from the following refinements:
a. Language edits from point 1 apply here too.
b. Specific intervention differences could be more precise (e.g., “intervention elements, such as duration, delivery, or behaviour change techniques”)
c. You say that sensitivity and subgroup analyses were conducted to assess heterogeneity – this is incorrect, since there were all separate analyses. Amend language to clarify.
d. Please include more detail on your methods, especially the month/year of last search (August 2024) and how you defined mindful movement interventions.
3. The rationale and relevant literature in the introduction need more detail and revisions for accuracy:
a. Be more specific in the first paragraph on challenges facing university students.
b. L56: you list mindfulness meditation as one of the types of physical activity, which is not true (because meditation is not a form of exercise). Revise if you mean mindful movement like yoga instead.
c. Provide better justification for the “distinct impacts of physical activity and mindfulness meditation” – at present you provide no evidence for this. It may be of interest to read a qualitative study we conducted on this – https://doi.org/10.3389/fpsyg.2022.984232 (disclaimer: this is my own work, I am not requiring you to cite it but it does provide relevant context).
d. Similarly in third paragraph, you are missing a relevant piece of research (review of interventions combining physical activity and mindfulness components for mental health – https://doi.org/10.1016/j.mhpa.2023.100575 ; same disclaimer as above). This could help you better communicate the rationale for reviewing mindful movement over either component separately. There are also relevant theoretical frameworks you could consider (e.g., Schuman-Olivier et al., 10.1097/HRP.0000000000000277)
e. L85: Good definition of mindful movement in your review but I suggest moving this higher up in the Introduction.
f. L90: Hypotheses are not required nor appropriate for a meta-analysis. Remove this.

Experimental design

4. The methods section also needs refinement and clarification.
a. Literature search: You can put the whole research strategy into supplementary materials or a table, if you are short on words for the main paper.
b. Participants section, L124: Make clearer the level of symptoms your sample had – did they all have clinical diagnoses, subclinical levels (e.g., threshold on validated scales), or did the studies just measure anxiety/depression in healthy samples? Why did you exclude physical activity education students?
c. Similarly, L124: What do you mean by primarily yoga interventions? What other types of mindful movement were there? The word primarily again leads to imprecise description of your method.
d. L159: While not being able to blind participants in behavioural trials is normal, simply excluding the category from risk of bias analyses is not appropriate – instead, the blinding category should be included and reported. You can explain this and comment on risk of bias beyond this category in the manuscript itself (if helpful, this review does the same – https://doi.org/10.1016/j.mhpa.2023.100575).
e. Were all your analyses specified and pre-registered before you conducted them?

Validity of the findings

5. Results section could also benefit from edits for clarity and accuracy:
a. L206: You say that outcomes were primarily from validated scaled, even though your methods section lists validated measures as an inclusion criterion – therefore all results must have come from validated measures? Please address this discrepancy.
b. Sensitivity analyses, L240: This is the first time we are hearing about sensitivity analyses being done. Include what you did for your sensitivity analysis and whether they were pre-registered in the Methods section, so it doesn’t come as a surprise to the reader.
c. L246: Which studies have greater influence on heterogeneity, and which study characteristics are you talking about here? Provide more details and examples.
d. L284: Begg’s and Begg’s corrected tests need to be explained in the Method and explained why they were done. The reader should not be surprised by any analysis done in the results.
e. L305: What do you mean by systematically omitting individual studies? How were they chosen and why?
f. L344: The adjusted R2 cannot be a negative value
g. Certainty of evidence section: Are you using a framework for these analyses? Again, explain them better in the Method so that the reader knows how to expect. This should include relevant citations of the framework and references where the reader can learn how to conduct these analyses themselves.
6. Discussion is your weakest section. It mostly just repeats the results without appropriately placing them in context of existing research or discussing the practical/clinical/research implications of your findings. I suggest you rewrite this section with a focus on those points. In addition:
a. Second paragraph (L403 onwards) is just a repetitive collection of sentences without much reference to your own work.
b. L403 you cite a non-peer reviewed trial registration (ChiCtr, 2022), which is not appropriate. Please provide a more suitable citation – there is plenty of peer reviewed work available on this topic.
c. L431: This is the first time you mention subgroup analyses by intervention duration – this is an interesting finding that should have been described in the Results first. The reader should not find out any new results in the Discussion section.
d. L454: You contradict yourself in your estimate of reliability of the meta-analysis’ findings. I suggest softening the language around the highly reliable conclusions, since lack of evidence for publication bias is not the only factor in reliability of results (e.g., your inclusion criteria, rigour of bias assessments, and comprehensiveness of search would also have implications).
e. I have concerns about your risk of bias assessment process: The Cochrane RoB tool has very clear instructions in that anything not reported is considered high risk (red category), so a study that does not report something cannot be considered overall low risk of bias. Your process description suggests you did not follow this. In Discussion (L461), you also say that lack of reporting of randomisation may have impacted your RoB assessment – in fact, the RoB tool clearly states that if randomisation is not reported, the study is considered red in that category and overall cannot be more than moderate risk. Please consult the Cochrane RoB tool instructions and resources, and revise your assessments and conclusions accordingly.

·

Basic reporting

See my comments.

Experimental design

See my comments.

Validity of the findings

See my comments.

Additional comments

Thanks for opportunity review revised manuscript entitled ‘‘Integrating mindfulness and physical activity: a meta-analysis of mindful movement interventions for anxiety and depression among university students’’ for Peerj journal. Authors examined effectiveness of mindful movement interventions for symptoms of anxiety and depression in Chinese language and English articles. As an experienced editor and reviewer, I think that although the article is generally well-written some comprehensive revisions must be made before publication of article. I can summarize main problems as follows section by section when possible with suggestions.
Title
1. Page 5, Line 1-3: The title of article must be revised to reflect symptoms of anxiety and depression as almost any of used instruments are diagnostic tools. One revision may be that ‘‘Integrating mindfulness and physical activity: a meta-analysis of mindful movement interventions for symptoms of anxiety and depression among university students’’ it is straightforward and clear.
Abstract
2. Page 6, Line 23-24: The first sentence of Background is overgeneralization and must be corrected ‘‘University students face significant mental health challenges due to employment pressures, unhealthy lifestyles.’’
3. Page 5, Line 27-28: Following sentence ‘‘This study evaluated the effects of mindful movement interventions on anxiety and depression among university students’’ must move Background section.
4. Page 5, Line 32: Following sentence must revise ‘ ‘Pairwise meta-analysis indicated that….’’ as ‘ ‘ The findings of pairwise meta-analysis indicated that….’’ I also think pairwise must remove it is unnecessary but scientifically correct.
5. Page 5, Line 33: Authors must provide long name of abbreviations in its first use. Following must correct as ‘ ‘SMD = -0.42, 95% CI: -0.52 to -0.31, P < 0.0001, I² = 70%)’’ as (Standardized Mean Difference [SMD] = -0.42, 95% CI = -0.52, -0.31, p < 0.0001, I² = 70%).
6. Manuscipt General: I also corrected reporting %95 confidence interval and p value reporting. Authors must correct these along the manuscript.
7. Page 5, Line 37: In the following ‘ ‘(P = 0.495).’’ p must small and italic.
8. Page 5, Line 39-41: Anthropomorphism exist in following sentence and must revise ‘ ‘Mindful movement effectively improves mental health among university students, with no significant differences in the effects of yoga, tai chi, and qigong on reducing anxiety and depression.’’ One revision may be that ‘ ‘Mindful movement interventions is associated with enhanced mental health among university students, with no significant differences in the effects of yoga, tai chi, and qigong on reducing symptoms of anxiety and depression.’’
9. Page 5, Line 45: The first letter of all keywords must be small.
Introduction
10. Page 6, 58-60: The first paragraph is without a conclusion ‘ ‘This raises a critical question: could mindful movement, which integrates physical activity with mindfulness meditation, provide more effective treatment outcomes for university students struggling with anxiety and depression?’’ So what?
11. Page 6, Line 72: Add a space between n and ( in the following ‘ ‘depression(Guo et al., 2020; Lin et al., 2022).’’
12.
13. Introduction, General: Because the dependent variable of study was psychological stress, Introduction section must begin with a general introduction of commonality of these symptoms among university students.
14. Introduction/Manuscipt, General: Each paragraph must consist of at least three at most eight sentences as per APA 7 rules. Authors must correct Introduction as well as along the manuscript as per this rule.
15. Introduction, General: The second problem of Introduction section is that authors must give information about importance of this study in their cultural context. Specifically, why it is important and necessary to examine effectiveness of mindful movement interventions for anxiety and depression among Chinese university students? This is an important reason for conducting this study. This information must give in introduction section at least one or two paragraph.
16. Introduction, General: The third problem of Introduction section is that authors must give findings of previous meta-analyses? What they found, they only give brief information on line 77-84. Moreover, what their limitations that necessitate this study?
17. Introduction, General: Authors must also give possible advantages of using mindful movement interventions among university students after Line 61-68.
Method
18. Method, General, Line 93: Survey methodology must rename as Method. Authors did not conduct a survey.
19. Method, General, Line 94-96: Research design title is completely missing and must be added and following must move there ‘‘This meta-analysis is reported in accordance with the PRISMA (Preferred Reporting Items Systematic Reviews and Meta-Analyses) guidelines (Page et al., 2021) and registered the study in PROSPERO (CRD42024569238).’’
20. Page 8, Line 104-119: Following must report in a table or move to supplementary material ‘ ‘The search strategy used in PubMed was as follows: ((((("Students"[Mesh]) OR ("Young Adult"[Mesh])) OR ((((university student[Title/Abstract]) OR (college student[Title/Abstract])) OR (undergraduate student[Title/Abstract])) OR (freshman[Title/Abstract]))) AND (((("Mindfulness"[Mesh]) OR ("Exercise"[Mesh])) OR ("Movement"[Mesh])) OR (((((((((Mindful Movement[Title/Abstract]) OR (Yoga[Title/Abstract])) OR (Tai Chi[Title/Abstract])) OR (Qigong[Title/Abstract])) OR
(Mindful Walking[Title/Abstract])) OR (Physical Activity[Title/Abstract])) OR (Aerobic
Exercise[Title/Abstract])) OR (Exercise Trainings[Title/Abstract])) OR (Resistance
Training[Title/Abstract])))) AND ((((((("Mental Health"[Mesh])) OR (((Health,
Mental[Title/Abstract]) OR (Mental Hygiene[Title/Abstract])) OR (Hygiene,
Mental[Title/Abstract]))) OR ("Anxiety"[Mesh])) OR ((((Angst[Title/ Abstract]) OR (Nervousness[Title/Abstract])) OR (Hypervigilance[Title/Abstract])) OR
(Anxiousness[Title/Abstract]))) OR ("Depression"[Mesh])) OR (((DepressiveSymptoms[Title/Abstract]) OR (Depressive Symptom[Title/Abstract])) OR (EmotionalDepression[Title/Abstract])))) AND (((Randomized controlled trial[Title/ Abstract]) OR(randomized[Title/Abstract])) OR (placebo[Title/Abstract])).’’
21. Page 8, Line 125-126: Following sentence ‘‘This study investigates the effects of Mindful Movement interventions on alleviating anxiety symptoms.’’ must correct as ‘‘This study investigates the effects of Mindful Movement interventions on alleviating symptoms of anxiety and depression.’’
22. Page 9, Line 162: we must revise as We
23. Page 9, Line 165: high ROB.(Cipriani et al., 2018) must revise as high ROB (Cipriani et al., 2018).
24. Methods, General: A lot of different standardized mean difference exist Cohen d hedges g which one specifically used? Please clearly indicate in Statistical analyses section.
25. Page 9, Line 180: Following must not be bbolg moreover p must be small P > 0.05
Results
26. Results section general: All small n representing subgroups must be italic.
27. Results section general: All p value must report with three decimals in the text and tables.
28. Results section general: All 95% CI: must report as 95% CI =
29. Results section general: Along the results section and along the manuscript all findings must report with two or three decimals consistently.
30. Results section general: All p values like this p = 0.000 must correct as p < 0.001
31. Results section general: Along the results section and along the manuscript add a space before and after =
32. Page 10, Line 197: Following only. (See Result map, Fig. 1) correct as only (See Result map, Fig. 1).
33. Page 10, Line 198: Remove following, I think it is a mistake: Fig. 1 Flow diagram of systematic literature search
34. Results section general: Writing of all subtitles must correct in Results section. For example, LITERATURE SELECTION must correct as Literature Selection
35. Page 10, Line 208-210: There is no findings to correct this right now and must remove/delete from this section. ‘‘These studies highlight the diversity of interventions and sample sizes, with consistent evidence of their positive effects on mental health outcomes.’’
36. Results section general: All number below 10 must report as four studies, one study …etc.
37. Page 11, Line 227-228: Remove following, I think it is a mistake: Fig. 2 Risk of bias graph Fig. Risk of bias summary
38. Page 11, Line 232: instead of improve, please use reduce ‘‘significantly improve anxiety levels…’’
39. Page 11, Line 234, 235: Please revise following ‘ ‘Moreover, the P-value (< 0.0001) demonstrates that the results significantly favor the mindfulness-based intervention group over the control group.’’ as ‘ ‘Moreover, the highly significant p-value (< 0.0001) indicates that the mindfulness-based intervention group showed a statistically significant improvement compared to the control group.’’ for better flow.
40. Page 11, Line 232: Move following end of sentence = (fig.4) as ….students (Fig. 4).
41. Results section general: do not use abbreviations for Table names or Figures. Figure 1, Table 1 is the correct from.
42. Page 11, Line 232: Remove following, I think it is a mistake: Fig. 4 The forest plot of anxiety-related outcomes
43. Results section general: When authors refer figures and tables using this format is preferable. As shown in Figure 6, the subgroup analyses indicated that…….
44. Page 11, Line 232: Remove following, I think it is a mistake: Fig. 6 Subgroup Analysis of Anxiety
45. Page 14, Line 232: Remove following, Fig. 11 Bias assessment funnel plot of depression
46. Page 14, Line 232: Following sentence must move from results, it can be used in Discussion section. ‘‘These findings highlight the need for more standardized intervention protocols and further research to address variability and strengthen the evidence base.’’
Discussion and others
47. Discussion general: No need subtitle to discussion section and must remove.
48. Discussion general: Practical implications are completely missing and must be added.
49. References, general: Issue number was missing in most references and must be added.
50. Table 2 and Table 4, Table 3 and Table 5 may be combined.
51. The image quality is low for some figures. Authors can use 400 dpi solutions.

---

## Round 0.2 · Minor Revisions

Formatting and Style
• Headings: Please ensure that the heading for the Abstract and the corresponding section in the main article is "Methods" not "Methodology". The main heading for presenting your findings within the article should be "Results".
• Case Consistency: Throughout the entire manuscript, "yoga," "tai chi," and "qigong" should be written in lowercase. They are currently capitalized in several places. Please be consistent in writing "tai chi" in full throughout the manuscript, rather than using the abbreviation "TCC".
• Font Consistency: All text currently in Arial font needs to be changed to Times New Roman.
• Spacing: There are a few inconsistencies with spacing that need to be corrected throughout the manuscript.

Content and Clarity
• Literature Search Details: In the section detailing the literature search ( line 143 of the tracked changes manuscript), please provide the initials of the researchers who conducted the literature search in brackets.
• Ambiguous Statement: The statement "the literature was considered to be deleted and not evaluated" (lines 287-288 in the tracked changes manuscript) is unclear and requires rephrasing to convey the intended meaning accurately.
• PRISMA Flow Diagram: For the PRISMA flow diagram, please adhere to the standard conventions for writing details for "records excluded", "reports not retrieved", and "reports excluded".
• Formatting and Proofreading:
• The paper would benefit from editing and proofreading.
• Please check for spacing issues throughout the manuscript. There are a few mistakes with spacing that need correction.
• All text currently written in Arial font must be changed to Times New Roman.
• Tables and Figures:
• There are too many tables and figures presented in the main body of the manuscript. which can make it appear cluttered. It is suggested to move some of these to a supplementary materials section.

·

Basic reporting

See my comments.

Experimental design

See my comments.

Validity of the findings

See my comments.

Additional comments

I am generally satisfied with revision. It can be published in this form.

---

## Round 0.3 · accepted · Accept

Thank you for your revised submission. I am happy to accept your paper for publication.